Immunogenicity evaluation of MS2 phage-mediated chimeric nanoparticle displaying an immunodominant B cell epitope of foot-and-mouth disease virus

Wang Guoqiang 1 2
Liu Yunchao 2
Feng Hua 2
Chen Yumei 3
Yang Suzhen 2
Wei Qiang 2
Wang Juan 4
Liu Dongmin 4
Zhang Gaiping zhanggaiping2003@163.com 1 2 5
1 College of Animal Science and Veterinary Medicine, Henan Agricultural University , Zhengzhou , China
2 Henan Provincial Key Laboratory of Animal Immunology, Henan Academy of Agricultural Sciences , Zhengzhou , China
3 School of Life Sciences, Zhengzhou University , Zhengzhou , China
4 Henan Zhongze Biological Engineering Co., Ltd , Zhengzhou , China
5 Jiangsu Co-Innovation Center for the Prevention and Control of Important Animal Infectious Disease and Zoonose, Yangzhou University , Yangzhou , China
Rosano Camillo
Electronic publication date: 2018 May 23
Publication date: 2018
Volume: 6
Electronic Location ID: e4823
Received 2018 Jan 16; Accepted 2018 May 2
Copyright: ©2018 Wang et al.
Copyright year: 2018
Copyright holder: Wang et al.
License: This is an open access article distributed under the terms of the Creative Commons Attribution License, which permits unrestricted use, distribution, reproduction and adaptation in any medium and for any purpose provided that it is properly attributed. For attribution, the original author(s), title, publication source (PeerJ) and either DOI or URL of the article must be cited.
License URL: https://creativecommons.org/licenses/by/4.0/

Keywords: Foot-and-mouth disease virus, Chimeric nanoparticles, MS2 bacteriophage, G-H loop

Funding: National Key R&D Program 2017YFD0501103 2016YFD0501503 Henan Province Industry-University-Research Collaboration Project 172107000037 This work was supported by grants from the National Key R&D Program (2017YFD0501103 and 2016YFD0501503) and the Henan Province Industry-University-Research Collaboration Project (172107000037). The funders had no role in study design, data collection and analysis, decision to publish, or preparation of the manuscript.

==============================
Foot-and-mouth disease (FMD) is a highly contagious disease of cloven-hoofed animals that has caused tremendous economic losses worldwide. In this study, we designed a chimeric nanoparticles (CNPs) vaccine that displays the predominant epitope of the serotype O foot-and-mouth disease virus (FMDV) VP1 131-160 on the surface of MS2 phage. The recombinant protein was expressed in Escherichia Coli and can self-assemble into CNPs with diameter at 25–30 nm in vitro. A tandem repeat peptide epitopes (TRE) was prepared as control. Mice were immunized with CNPs, TRE and commercialized synthetic peptide vaccines (PepVac), respectively. The ELISA results showed that CNPs stimulated a little higher specific antibody levels to PepVac, but was significantly higher than the TRE groups. Moreover, the results from specific IFN-γ responses and lymphocyte proliferation test indicated that CNP immunized mice exhibited significantly enhanced cellular immune response compared to TRE. These results suggested that the CNPs constructed in current study could be a potential alternative vaccine in future FMDV control.

Introduction

Foot-and-mouth disease virus (FMDV) infects over 70 species of cloven-hoofed animals and has caused enormous economic losses to stockbreeding industry worldwide (Diaz-San Segundo et al., 2017; Pereira, 1976). Conventional inactivated FMDV vaccine has been widely used and played a crucial role in epidemic control and eradication of FMDV globally. However, it remains some problems of the inactivated FMDV vaccine, such as the risk of virus escape during vaccine production and difficulty in differentiating infected from vaccinated animals, which limit the application of inactivated FMDV vaccine in emergency control programs (Dong et al., 2015; Parida, 2009; Wang et al., 2002). Epitope-based polypeptide vaccines are well known for their superior abilities to provide more effective antigens and it could be effectively distinguished between infected and vaccinated animals. However, limited cellular immune response and immune protection were induced by these epitope vaccines in large host animals, resulting in the failure of FMDV prevention (Rodriguez et al., 2003; Taboga et al., 1997).

Neutralizing epitopes of FMDV, which distributed in structural proteins VP1, VP2 and VP3, is critical to an effective immune protection. FMDV VP1 G-H loop has been identified as a primary antigenic epitope, which can effectively induce FMDV specific neutralizing antibodies (Bittle et al., 1982; DiMarchi et al., 1986). The highly conserved arginine-glycine-aspartic acid (RGD) motif is located on the top of G-H loop of VP1. FMDV can use RGD-binding integrins as receptors to initiate infection (Burman et al., 2006). Therefore, based on the dominant epitope G-H loop, a large number of FMD epitope vaccines are being studied. A commercial FMD synthetic-peptide vaccine (UBITh® vaccine) containing an entire G-H loop domain and extensive flanking sequences (129-169) was licensed for use in China (Cao, Lu & Liu, 2016).

Nanoparticles-based antigen display technology provides an approach to improving the immune response and immune protection effect of subunit vaccines (Chackerian, 2007; Crisci et al., 2009; Crisci et al., 2012; Xu et al., 2017). MS2 phage is a novel display and delivery platform for foreign peptide epitopes. MS2 phage belongs to the Leviviridae family with small positive-sense single-stranded RNA bacteriophages, which is encapsulated by an icosahedral capsid comprised of 180 copies of coat protein (CP) and a single copy of mature protein (AP) (Koning et al., 2016; Wei et al., 2008). The CP of MS2 phage can self-assemble to form virus-like particles (VLPs) in Escherichia Coli (E. coli) in the absence of viral RNA (Caldeira & Peabody, 2011; Lino, Caldeira & Peabody, 2017). The AB loop of CP is exposed on the surface of the phage particles, which tolerates exogenous peptides insertion without affecting the self-assembly of CP (Fu & Li, 2016; Mastico, Talbot & Stockley, 1993). These characteristics play an important fundamental basis for remodeled epitope peptide to enhance the efficacy of vaccines.

In the present research, we inserted the epitope sequence (131-160, including G-H loop and its flanking sequence) into the AB loop of MS2 phage CP. Chimeric protein was expressed in E. coli and self-assembled into chimeric nanoparticles (CNPs). The immunoassay test showed CNPs could efficiently enhance the antibody levels and cellular immune response compared to tandem repeat peptide epitopes (TRE) and commercialized synthetic peptide vaccine (PepVac) groups. Together, our results suggested that MS2-mediated CNPs provide a favorable platform for displaying foreign epitopes and an innovative approach to develop alternative vaccines for FMDV.

Materials and Methods

Animal, commercial vaccine and antibody

Twenty six-week-old female Kunming mice were provided by Laboratory Animal Center, Zhengzhou University. This study was performed with the approval of the Animal Experiment Committee of Henan Academy of Agricultural Sciences (Approval number SYXK 2014-0007). All animals used in this study were humanely maintained and euthanized according to the animal ethics procedures and guidelines of China.

Commercialized synthetic peptide vaccine (peptide 2600+2700+2800) (PepVac) was purchased from Shanghai Shen-Lian Biomedical Corporation (Shanghai, China; http://www.shenlianbiotech.com.cn/product-4.html). Polypeptide 2600, 2700 and 2800 represent the epitope sequences of pandemic strains of ME-SA topotypes, CATHAY topotypes and SEA topotypes of serotype O FMDV, respectively.

The monoclonal antibodies against FMDV VP1 G-H loop (141-160) was prepared and provided by the Key Laboratory of Animal Immunology, Henan Academy of Agricultural Sciences, which was generated using hybridoma technology as described in previous reference (Ma et al., 2010). Binary ethylenimine (BEI) inactivated FMDV, guinea pigs anti-FMDV/O hyper-immune serum and rabbit polyclonal antibodies against FMDV were obtained from the National FMD Reference Laboratory of the People’s Republic of China. Horseradish peroxidase (HRP)-conjugated goat anti guinea pigs IgG was obtained from Sigma, and Horseradish peroxidase (HRP)-conjugated goat anti mouse IgG was obtained from Abcam.

Table 1 Sequences of primers for PCR/OE-PCR.

Primer name	Primer sequence (5′-3′)	
MS2-F	CGggatccGTGCGAGCTTTTAGTACCCTTGA	
MS2-R	CCCaagcttTGTTGTCTTCGACATGGGTAATCCTC	
IN-F	CTGACCAACGTGCGTGGCGATCTGCAAGTCCTGGCACAGAAAGCTGCACGTCCTCTGCCTACTGGCGACGTGACTGTCGCCCCAAGCAA	
IN-R	TGCCAGGACTTGCAGATCGCCACGCACGTTGGTCAGGGAACCCTCAGCGTACTTGCAGTTCCCGTTTCCGCCATTGTCGACGAGAACGAAC	
Notes.

Lowercase part in primers MS2-F and MS2-R represent for restriction enzymes (Bam H I and Hin d III).

The sequences in bold type in two primers IN-F and IN-R were the epitope sequence 131–160 of VP1, and underlined 36 nucleotides were reverse complementary.

Figure 1 Schematic drawing of constructing process for recombinant plasmids.

(A and C) Schematic diagram of recombinant plasmid pCP and p-(EP131-160)3. (B) Strategy to construct recombinant plasmid pCP-EP131-160.

Plasmid construction

All the primers used to construct the recombinant plasmids were shown in Table 1. The restriction sites BamH I and Hind III were represented in lowercase in primers MS2-F and MS2-R. The sequences in bold type in two primers IN-F and IN-R were the epitope sequence 131–160 of VP1, and underlined 36 nucleotides were reverse complementary. The epitope sequence 131-1-60 came from the O/BY/CHA/2010 strain (GenBank: AET43040.1). The cDNA of encoding AP and CP sequences of MS2 were reversely transcribed from MS2 genome mRNAs (purchased from Sigma) by primers MS2-F and MS2-R using a One-Step RT-PCR Kit (TaKaRa, Shiga, Japan) and cloned into T-Vector pMD™ 19 (Simple). The recombinant plasmid was named as T-MS2. The construction diagrams of recombinant plasmids were depicted in Fig. 1. Fragment including AP and CP sequences was amplified from T-MS2 plasmid using primers MS2-F and MS2-R and cloned into pET28a plasmid named as pCP. The whole chimeric sequence was amplified by overlap extension PCR (OE-PCR) (Liu et al., 2006; Yang et al., 2005). Briefly, the upstream and downstream fragments were amplified from T-MS2 plasmid using MS2-F/IN-R and MS2-R/IN-F primer pairs, respectively. The mixture of equimolar upstream fragment and downstream fragment was used as template, and then the whole chimeric sequences containing AP gene, CP gene and the epitope sequence 131-160 were amplified using primers MS2-F and MS2-R and cloned into pET28a vector. The recombinant plasmid was named as pCP-EP131-160. The tandem repeat sequence consisting of three copies of the epitope sequence 131-160 connected by GSGSGS was artificially synthesized and cloned into pET28a vector named as p-(EP131-160)3. Recombinant plasmids pCP and pCP-EP131-160 were identified by digestion with restriction enzymes BamH I and Hind III, respectively, and p-(EP131-160)3 was identified by restriction enzymes Nde I and Xol I. Finally, all the recombinant plasmids were identified by sequencing.

Recombinant protein expression and purification

Recombinant plasmids pCP, pCP-EP131-160 and p-(EP131-160)3 were transformed into E. Coli BL21 (DE3), respectively. A single clone was selected from LB agar plate and cultured in LB medium supplemented with 50 µg/ml kanamycin. Until the OD600 reached 0.8, target proteins were induced by 0.3 mM isopropyl β-D-thiogalactoside (IPTG). After induction at 20 °C for 16 h, the cells were harvested by centrifugation at 6,000 × g for 15 min at 4 °C. The cell pellet was re-suspended in PBS buffer and sonicated seven times for 30 s each on ice, and then was centrifuged at 12,000 × g for 15 min at 4 °C. The soluble fraction in supernatant and insoluble fraction in precipitation were analyzed using SDS-PAGE and Western blotting. The soluble CP and chimeric protein in supernatant were purified separately as below: DNase I and RNase A with a final concentration of 1 µg/ml were added into the supernatant at room temperature for 30 min, respectively. Then solid NaCl with a final concentration 1 mol/L was added and incubated on ice for 1 h. After centrifugation at 11,000 × g for 10 min, PEG8000 was added into supernatant to a final concentration of 10% (w/v) and stored the mixture for at least 1 h. After centrifuged again, the pellet was re-suspended in PBS buffer. Followed by incubated with an equal volume of chloroform and then gently vortexed the mixture for 30 s, finally the aqueous phases containing chimeric protein were collected by centrifuged at 5,000 × g for 10 min. The chimeric protein was further purified by gel filtration chromatography (Capto Core 700; GE, Boston, MA, USA). Briefly, preliminary purified chimeric protein was pumped onto PBS buffer equilibrated Capto Core 700 column and the effluent containing target protein was collected directly.

Besides, the tandem repeat peptide epitopes (TRE) was purified by Ni-NTA column (Merck, Darmstadt, Germany). TRE was expressed as inclusion body (IB), so IB was dissolved in 8M urea and loaded onto Ni-NTA column equilibrated with 0.05 M carbonate buffer (pH = 9.0) containing 8M urea. After washing with 10 beds of carbonate buffer (50 mM imidazole, 8 M urea), the IB was eluted with carbonate buffer (100 mM imidazole, 8 M urea). Then the purified protein was gradient dialyzed with 0.05 M carbonate buffer (pH = 9.0) containing continuously decreased urea concentration (from 8 to 0 M) for 72 h. The concentration of purified recombinant proteins was calculated using Micro BCA™ protein assay kit (Thermo Scientific, Waltham, MA, USA) following the manufacturer’s protocol.

Identification of recombinant proteins

The purified chimeric protein was further characterized by transmission electronic microscopy (TEM) using the negative staining and particle size was analyzed by dynamic light scattering (DLS) as described before (Chandramouli et al., 2013).

The reactivity of purified recombinant proteins was analyzed by Dot-ELISA. Purified CP, CNPs and TRE were blotted onto a nitrocellulose (NC) membrane. Inactivated FMDV was used as positive control. The NC membrane was blocked with 5% skimmed milk for 2 h at 37 °C, and then incubated with guinea pigs anti-FMDV/O hyper-immune serum or anti-FMDV VP1 G-H loop (141-160) monoclonal antibodies as the primary antibodies, followed by a HRP-conjugated goat anti guinea pigs IgG or mouse IgG as secondary antibody. The Dot-ELISA was visualized using 3-amino-9-ethylcarbazole (AEC) (ZSGB-BIO, Beijing, China), which is a substrate of peroxidase.

Vaccine preparation and immunization

The purified CNPs and TRE were emulsified with adjuvant Montanide ISA 50V2 (Seppic, France) for animal vaccination, respectively. The ratio of aqueous antigen to the oil adjuvant was 1:1 (V/V). Twenty female Kunming mice were randomly divided into four groups (five each group). Groups 1 and 2 were vaccinated subcutaneously with 15 µg of CNPs and 30 µg of TRE, respectively. Group 3 was inoculated with 100 µl of PepVac (more than 7.5 µg) used as positive control, and Group 4 was vaccinated of 100 µl of PBS with the same volume of adjuvant as negative control. All mice received boost vaccination at 28 days after first immunization.

Detection of anti-FMDV-specific antibodies

Serum samples were collected weekly from the tail vein after the first immunization until the eighth week. All the samples were tested for anti-FMDV specific antibodies by ELISA. Briefly, 100 µl per well rabbit polyclonal antibody against FMDV was coated on 96-well ELISA plate with and incubated at 4°C overnight. The 96-well plate was blocked with 250 µl PBST including 5% skimmed milk at 37 °C for 2 h, then washed three times with PBST. 20-fold diluted inactivated FMDV was added to the 96 plate with 100 µl per well and incubated at 37 °C for 1 h. After being washed with PBST, the serum samples with a dilution of 1:100 were added and incubated at room temperature for 1 h. Then the plate was reacted with HRP-conjugated goat anti mouse IgG at 37 °C for 1 h. Finally, after being washed five times, the reaction substrate was respectively added to each well and incubated at 37 °C for 10 min. Then, the reaction was stopped by 2M H2SO4, and the OD450 nm values were measured by a spectrophotometer.

Spleen lymphocyte proliferation assay

The spleen lymphocytes were isolated from immunized mice at 28 days after booster immunization by using a lymphocyte separation kit (Solarbio, Beijing, China). The Cell Counting Kit-8 (CCK-8) (Solarbio, Beijing, China) was used to detect cell proliferation according to manufacture instruction. (2-(2-Methoxy-4-nitrophenyl)-3- (4-nitrophenyl)- 5-(2,4-disulfobenzene)-2H-tetrazole monosodium salt) (WST-8) is the main component of the CCK-8 kit. Briefly, The spleen lymphocytes were re-suspended in RPMI-1640 medium containing 10% FBS, and incubated in triplicate in 96-well plate with a density of 5 × 105 cells/well at 37 °C for 24 h. Then, the cells were stimulated with 50 µl of inactivated FMDV (20 µg/ml). Concanavalin A (ConA, 5µg/ml) and unstimulated wells were used as the positive control and negative control. After incubation at 37 °C for 48 h, WST-8 (10 µl/well) was added to each well and incubated at 37 °C for 1 h. The absorbance of each well was measured at 450 nm. T lymphocyte proliferation was expressed as the stimulation index (SI), which was the ratio of OD450 nm of stimulated wells to OD450 nm of unstimulated ones.

Cytokines detection

Splenic lymphocytes culture supernatants stimulated with inactivated FMDV for 48 h were collected for evaluating the concentration of IL-2, IL-4 and IFN-γ. The assay and data calculation were performed by the commercially available ELISA kit (Bogoo, Shanghai, China) following manufacturers’ instructions.

Statistical analysis

Statistical analysis was performed by using GraphPad Prism 7.0 software. Two-way ANOVA and t-test method were employed for significant test. Dates were shown as the mean ±SEM, and P-values less than 0.05 was considered as statistically significant.

Results

Construction of recombinant vectors

A schematic diagram of obtaining the target gene and constructing three recombinant plasmids is outlined in Fig. 1. The MS2 (AP + CP) gene, upstream fragment, downstream fragment and the whole length chimeric genes were amplified by PCR/OE-PCR with the expected molecular size of 1,644 bp, 1,260 bp, 487 bp and 1,734 bp (Fig. 2A), suggesting that the epitope 131-160 gene was successfully inserted into CP of MS2. Recombinant plasmids, pCP, pCP-EP131-160 and p-(EP131-160)3, were also confirmed by restriction digestion (Fig. 2B) and sequencing (Shown in the Supplemental Information). The fragment sizes digested from recombinant plasmids pCP, pCP-EP131-160 and p-(EP131-160)3 were 1,644 bp, 1,734 bp and 306 bp, respectively, which were consistent with expected.

Figure 2 Amplification of target gene and identification of recombinant plasmids.

(A) PCR amplification products analysis using agarose gel. Lane 1, DL2000 DNA ladder; Lane 2, Mature protein (AP) and coat protein (CP) genes of MS2 showing 1,644 bp amplicon; Lane 3, Upstream fragment showing 1,260 bp amplicon; Lane 4, Downstream fragment showing 487 bp amplicon; Lane 5, The whole fragment (AP + CP + EP131-160) showing 1,734 bp amplicon. (B) The identification of recombinant plasmids by digestion with restriction enzymes. Lane 1, DL2000 DNA ladder; Lane 2–3, Recombinant plasmids pCP and pCP-EP131-160 were digested using restriction enzymes BamH I and Hind III, respectively; Lane 4, Recombinant plasmid p-(EP131-160)3 was digested by Nde I and Xol I.

Expression and purification of recombinant proteins

To further investigate whether the expressed recombinant proteins are soluble in solution, SDS-PAGE and Western blotting were performed in our studies. The result of SDS-PAGE indicated that approximately 50% of CP (16 kDa) and chimeric proteins (18 kDa) were expressed in soluble form, but TRE (10 kDa) was expressed as IB (Fig. 3A). Additionally, the recombinant proteins of the chimeric protein and TRE could be recognized by the anti-FMDV VP1 G-H loop (141-160) monoclonal antibodies, while CP did not show the similar phenomenon (Fig. 3B).The purities of three recombinant proteins were estimated to be over 85% (Fig. 3C).

Figure 3 Expression and purification of recombinant proteins.

(A) SDS-PAGE analysis of the solubility of three recombinant proteins. Lane 1, protein ladder; Lane 2–3, level of coat protein of MS2 expressed in soluble and insoluble fractions, respectively; Lane 4–5, level of chimeric protein expressed in soluble and insoluble fractions, respectively; Lane 6–7, level of tandem repeat peptide epitopes (TRE) expressed in soluble and insoluble fractions, respectively. As shown, coat protein of MS2, chimeric protein and TRE were detected with a molecule weight near 16 kDa, 18 kDa and 10 kDa, respectively. (B) Western blot analysis of recombinant proteins with anti-FMDV VP1 G-H loop monoclonal antibody. Lane 1–7, the order is consistent with SDS-PAGE (A). (C) SDS-PAGE analysis of the purity of three recombinant proteins. Lane 1, protein ladder; Lane 2, purified coat protein of MS2; Lane 3, purified chimeric protein; Lane 4, purified TRE.

Reactivity of CNPs and TRE

The reactivity of recombinant proteins was analyzed by Dot-ELISA. The results showed that CNPs and TRE could efficiently react with anti-FMDV hyper-immune serum (Fig. 4A) and anti-FMDV VP1 G-H loop (141-160) monoclonal antibodies (Fig. 4B), while CP did not show the similar phenomenon. These results suggested that the epitope sequence 131-160 was correctly displayed on the surface of A-B loop of MS2, and CNPs and TRE had a superior immune reactivity with anti-FMDV hyper-immune serum.

Figure 4 Immune-activity analysis of recombinant proteins.

Dot-ELISA immune assay with guinea pig anti-FMDV hyper-immune serum. (B) Dot-ELISA immune assay with anti-FMDV VP1 G-H loop monoclonal antibody. 1, Inactivated FMDV; 2, purified coat protein of MS2; 3, purified chimeric nanoparticles; 4, purified TRE.

Physical characterization of CNPs

To verify the self-assembly of chimeric protein into nanoparticles in vitro, the purified CNPs was analyzed by TEM. Results showed under pH 8.0 in 150 mM NaCl the purified chimeric proteins were assembled into nanoparticles with a diameter of 25–30 nm (Fig. 5A), which were consistent to the results of DLS (Fig. 5B).

Figure 5 Physical characterization of chimeric nanoparticles (CNPs).

(A) Transmission electron microscope (TEM) image of negative staining CNPs. The scale is 100 nm. (B) Dynamic light scattering results of CNPs.

Figure 6 CNPs specific antibody levels in mice.

Groups of mice were immunized with 15 µg chimeric nanoparticles, 30 µg tandem repeat peptide epitopes, commercialized synthetic peptide vaccine (more than 7.5 µg) and PBS. Sera were collected at every week after the first immunization for antibody levels until the eighth week. Date are presented as the mean ± SEM. Statistical differences were indicated by asterisks (* P < 0.05, **** P < 0.0001).

Antibody induction in immunized mice

To evaluate the immunogenicity of CNPs and TRE in vivo, anti-FMDV antibody titer in sera samples was evaluated by ELISA. As shown in Fig. 6, the antibody titers were increased in a time-dependent way and specific antibodies against inactivated virus could be detected at 14 days post vaccination (dpv) in the experimental group and the PepVac group. On day 28 dpv and 56 dpv, the CNPs group revealed the highest antibody levels, but no significant difference was found between the CNPs group and the PepVac group (P > 0.05). Mice immunized with CNPs induced significantly higher antibody titer compared to the TRE group (P < 0.01) and the PBS group (P < 0.0001). After booster immunization, antibody titers were further increased significantly except in the PBS group. Together, the CNPs displaying epitope 131-160 of FMDV showed the ability to induce a strong specific anti-FMDV humoral immune response compared to other groups.

T lymphocyte proliferation

The spleen lymphocytes were isolated from mice at 28 days after booster immunization and stimulated in vitro with inactivated FMDV. As shown in Fig. 7, the specific lymphocyte response levels of the CNPs, PepVac and TRE groups were significantly enhanced compared to the PBS groups (P < 0.01). The group of CNPs elicited higher lymphocyte proliferation responses than the TRE group (P < 0.05), while no significant differences were observed between the CNPs and PepVac groups (P > 0.05).

Figure 7 T-lymphocyte proliferation in mice.

Splenic Lymphocytes were isolated at 56 days post vaccination and stimulated with inactivated FMDV and concanavalin (CoA), respectively. Lymphocyte proliferation was analyzed using the WST-8 colorimetric assay. The stimulation index (SI) means the ratio of stimulated sample to unstimulated sample at OD450 nm. Statistical differences were indicated by asterisk (* P < 0.05, ** P < 0.01, *** P < 0.001).

Figure 8 Analysis of cytokines secreted by lymphocyte of mice.

Splenic Lymphocytes were stimulated with inactivated FMDV for 48 h. The culture supernatants were collected and tested by ELISA. (A–C) are the concentrations (pg/mL) of IFN-γ, IL-2 and IL-4 in the supernatants, respectively. Data are shown as mean ± SEM. Statistical differences were indicated by asterisk (* P < 0.05, ** P < 0.01, *** P < 0.001).

Cytokine assay

As it has been well known that the ability of lymphocytes to secret cytokines after stimulation is positively correlated with their functions. Thus, it is necessary to detect the cytokine levels after the stimulation. To assess the cytokine secretion of spleen lymphocytes after stimulation, IFN-γ, IL-2 and IL-4 concentrations in culture supernatants were evaluated by ELISA. As shown in Fig. 8, the CNPs group and the PepVac group induced greater IFN-γ and IL-2 levels than PBS group (Figs. 8A, 8B). However, there are no significant differences in the IL-4 level among three groups (P > 0.05) (Fig. 8C). Notably, the CNPs immunized group produced significantly higher IFN-γ levels than the TRE group and the PepVac group (p < 0.05). Taken together, the data suggested that CNPs immunization could enhance T lymphocytes immune response by elevated proliferation rate and cytokines secretion.

Discussion

FMDV remains as the main challenge in cloven-hoofed animals breeding worldwide (Sobrino et al., 2001), and innovative approach to produce safe and effective vaccine to prevent the occurrence and spread of FMDV is urgently needed. Infectious bursal disease subviral particles and porcine parvovirus subviral particles have been demonstrated to serve as an effective delivery and display platform for FMDV epitopes (Pan et al., 2016; Remond et al., 2009). However, these chimeric particles were obtained through eukaryotic cells, which limit the application of promotion in practice. Conveniently, the CP of MS2 phage could be expressed and assembled in E.coli, which makes it possible to achieve a rapid and low-cost production. Previous studies have shown that MS2 phage is a favorable platform for displaying and delivering epitope peptides (Fu & Li, 2016; Heal et al., 1999; Lino, Caldeira & Peabody, 2017). However, the limited tolerance of MS2 phage to this insertion and the misfolded, aggregated or degraded proteins resulting from long amino acids insertion remained to be the main problems (Caldeira & Peabody, 2011; Peabody, 1997). In this study, 30 amino acids polypeptide which came from the predominant epitope of FMDV were inserted into the coat protein of MS2 phage. Chimeric protein was expressed in E. coli, and can efficiently self-assembl into nanoparticles with a diameter of 25–30 nm (Fig. 5). Moreover, the epitope sequence 131-160 of VP1 was displayed on the surface of MS2, which was confirmed by DOT-ELISA (Fig. 4). These results indicated that the insertion of 30 amino acids did not affect the folding and assembly of coat protein.

The FMDV VP1 G-H loop (141-160) is the main immunogenic epitopes for inducing neutralizing antibodies (Morgan & Moore, 1990; Ochoa et al., 2000). MS2 mediated VLPs vaccine displaying G-H loop (141-160)of FMDV provides 65% protection against FMDV in guinea pigs and 60% protection against FMDV in pigs (Dong et al., 2015), which did not offer full protection. The flanking sequences of G-H loop can markedly strengthen its immune response (Dong et al., 2015; Fang et al., 2015). Moreover, the G-H loop of FMDV is an annular spatial conformation on the surface of natural virus particle, and correct conformation is crucial to induce effective humoral and cellular immune responses (Acharya et al., 1990; Cao et al., 2017; Rowlands et al., 1994). In the present research, we chose the epitope sequence 131-160 that contained G-H loop domain (141-160) and extensive flanking sequences to insert the AB loop of CP of MS2. Also, the epitope sequence 131-160 may still retain the circular structure through the AB loop of MS2. In fact, we had also attempted to insert epitope 124-167 of VP1 into the CP of MS2. But only a few chimeric proteins were obtained, which could not be assembled into nanoparticles (see Supplemental Information). This may be because insertions of 44 amino acids affected the CP of MS2 self-assembly. A total of 15 µg of CNPs stimulated a more enhanced humoral immune response than 30 µg of TRE (P < 0.01, Fig. 6), and induced slightly higher antibody levels compared to Pepvac, although there was no significant difference (P > 0.05, Fig. 6). These results indicated that the predominant epitope sequence 131–160 was displayed on the surface of MS2 at a high density, which stimulated stronger antibody levels in mice.

Based on the G-H loop and C-terminal sequence (200-213) of VP1, a variety of epitope vaccines have been developed to elicit high neutralizing antibodies titers in small animals, such as mice and guinea pigs (Su et al., 2007). However, several studies observed limited antibody levels and immune protection in host animals that induced by these epitope vaccines, which may be due to the lack of appropriate T-helper cell epitopes and low molecular weight of peptides (Cao, Lu & Liu, 2016; Rodriguez et al., 2003). CNPs, a special form of VLPs, have the capability of inducing extensive cell-mediated immune responses (Fu & Li, 2016; Ong, Tan & Ho, 2017). Notably, we observed elevated lymphocyte proliferation response in CNPs immunized mice than the TRE group (P < 0.05, Fig. 7). Moreover, the secretion of IFN-γ in the CNPs group was significant enhanced compared to the TRE group and the PepVac group (p < 0.05, Figs. 8A, 8B) and the IL -2 levels in the CNPs group was higher than the TRE group (p < 0.05, Fig. 8B). It is well known that cytokine IFN-γ and IL -2 are associated with cell-mediated immunity. Together with those results, we concluded that CNPs could elicit enhanced cell immune response compared to TRE and PepVac. Due to species differences, the following experiments will be performed to evaluate the immunogenicity and immuno-protection of CNPs in host animals.

In recent years, the purification processes of macromolecules, such as virus or VLPs, are mainly density gradient centrifugation and gel filtration chromatography. However, expensive and complicated purification processes limit the large-scale production of virus-like particles. (Dong et al., 2015; Liu et al., 2017; Pan et al., 2016). The new media Capto Core 700, with the capable of size separation and capture of small molecules, is designed for purification of viruses and other large biomolecules. In this study, we used gel filtration chromatography (Capto Core 700) for CNPs purification. Without a flow rate limit and no repeated elution, the effluent containing CNPs could be collected directly. More importantly, the purity efficiency of CNPs and CP was over 85%.

Conclusions

In conclusion, we have developed an MS2 phage mediated CNPs, with displaying predominant epitope 131–160 of VP1 on the CP of MS2 phage, which could be expressed and self-assembled into nanoparticles in E.coli. In addition, the CNPs had improved immunogenicity compared with PepVac and TRE, which could elicit enhanced specific anti-FMDV antibody titers and elevated cellular immune response compared to TRE in mice. Therefore, MS2 phage-mediated CNPs elicit an effective humoral and cellular immune response in mice, which provides a new perspective for future studies to subunit vaccines.

Supplemental Information

Supplemental Information 1 Raw data of ELISA to detect specific antibodies

Mice were immunized with PepVac, CNPs, TRE and PBS respectively. Mice sera were collected every week after immunization and used to detect the specific antibodies against inactivated FMDV by ELISA.

Click here for additional data file.

Supplemental Information 2 Raw data of lymphocyte proliferation assay

The T-lymphocyte proliferation of mice immunized with synthetic peptide vaccine, CNPs, TRE and PBS on 56 dpv. SI: OD450nm value of culture with FMDV stimulation/OD450nm value of culture without stimulation.

Click here for additional data file.

Supplemental Information 3 Raw data of ELISA to detect cytokines concentrations

Splenic Lymphocytes were stimulated with inactivated FMDV for 48 h. The culture supernatants were collected, and the concentrations (pg/mL) of IFN- γ, IL-2 and IL-4 in the supernatants were tested by ELISA following manufacturers’ instructions.

Click here for additional data file.

Supplemental Information 4 Recombinant plasmids sequencing results

Recombinant plasmids, pCP, pCP-EP131-160 and p-(EP 131-160)3, were confirmed by sequencing.

Click here for additional data file.

Figure S1 Transmission electronic microscopy (TEM) figure of chimeric protein

The sequence of 124-167 was inserted into the CP of MS2. But only a few chimeric proteins were obtained, which could not be assembled into nanoparticles. TEM picture as shown.

Click here for additional data file.

Additional Information and Declarations

Competing Interests

Author Contributions

Animal Ethics

Data Availability

Juan Wang and Dongmin Liu are employees of Henan Zhongze Biological Engineering Co., Ltd.

Guoqiang Wang performed the experiments, prepared figures and/or tables, authored or reviewed drafts of the paper, approved the final draft.

Yunchao Liu conceived and designed the experiments.

Hua Feng, Suzhen Yang, Juan Wang and Dongmin Liu contributed reagents/materials/analysis tools.

Yumei Chen and Qiang Wei analyzed the data.

Gaiping Zhang conceived and designed the experiments, approved the final draft.

The following information was supplied relating to ethical approvals (i.e., approving body and any reference numbers):

This study was performed with the approval of the Animal Experiment Committee of Henan Academy of Agricultural Sciences (Approval number SYXK 2014-0007).

The following information was supplied regarding data availability:

The raw data are provided in the Supplemental Files.

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
