# Peer review of "Immunogenicity evaluation of MS2 phage-mediated chimeric nanoparticle displaying an immunodominant B cell epitope of foot-and-mouth disease virus"

_PeerJ, doi:10.7717/peerj.4823_

## Round 0.1 · original submission · Major Revisions

The manuscript adds very little to the existing literature. However the referees made a great job suggesting many possibilities to improve the paper.

·

Basic reporting

Although generally adequate the quality of the English could benefit from help from a native English speaker.
Terminology could be improved. For example the full details of the FMDV used should be given and the sequence that was used should be referred to as the "FMDV VP1 G-H loop" and this used throughout.
What is PepVac? I could find no information on this. The authors imply that it is a vaccine that is in commercial use - is this the case or is it a synthetic peptide that was made under contract?
What was the template used for amplification of the VP1 G-H loop sequence? Presumably FMDV. If so, what serotype and what was the sequence?
Line 85 - how many repeats were linked?
Line 96 - how much 1M NaCl was used?
Line 117 - what about PepVac?
Line 127 - How much antigen does 100ul of PepVac contain and what is the antigen?
Line 148 - what is WST-8? Define.
Line 167 - Which restriction enzymes were used and where did they cut?
Fig 1 legend - What are upstream and downstream fragments and where is whole fragment? What is SOE-PCR product? What is Fig 1B? What was used for digestion?
I found the figure legends generally unsatisfactory.
There should be a diagram of the cloning strategy.
Fig 2 legend - what is meant exactly by lysate supernatant and lysate precipitate?
Fig 3 - there are no controls. Unmodified MS2 particles and an irrelevant peptide for example.
Fig 4 - the particles should be shown at a higher magnification.
Fig 5 - the responses look quite similar at 1/20. Were dilution series tested to get a better idea of the relative responses? Did the boost do anything? - what happens is they are not boosted? Do the titres still rise?

Experimental design

The experimental designs are appropriate for the aims of the study but much clearer and more detailed descriptions should be given (see 'basic reporting' section).
However, it has been known for decades that immune responses to peptides can be improved by linking sequences designed to provide T cell help. Why was this not done with the tandem repeat construct?
It has also been known for many years that the ability of peptide vaccines to induce protective responses is species specific and small laboratory animals are far easier to protect than commercial target species. No attempt was made to test or address this problem.

Validity of the findings

This study differs little from the observations publish by Dong et al (as referenced) who also made chimeric MS2 VLPs incorporating sequence of the FMDV VP1 G-H loop. The authors should describe how their study adds to the already published literature.

Reviewer 2 ·

Basic reporting

-

Experimental design

-

Validity of the findings

-

Additional comments

Wang et al. report a potential chimeric nanoparticle-based vaccine (CNP) against the Foot-and-Mouth disease, which is a viral infectious disease of cloven-hoofed animals. They generated the CNP vaccine, and analyzed its immunological responses in mice, in comparison with other type of vaccines, such as, a commercialized vaccine PepVac. I have come comments, below:

1. In general, an abbreviation should be used after its full name/word has been mentioned. For example, in the abstract, should FMDV stand for the Foot-and-Mouth virus? Although, a full name/word and its corresponding abbreviation have already been mentioned in the abstract, when they come into the text (i.e., in the introduction part), should they be mentioned again in full name/word before its abbreviation could be used? Please check the use of abbreviations throughout the manuscript. For example, Line 55, what does 'AP' stand for? Line 62-64, CNPs, TRE, and PepVac stand for? Line 122, AEC?

2. The first sentence of each paragraph should be indented.

3. Please have a native English speaker proofread the manuscript to ensure clarification and grammar of the text.

4. Line 69, should the total number of mice be mentioned here?

5. Line 74, for those who may not be familiar with this type of vaccine, what is the meaning of 'peptide 2600+2700+2800'?

6. Line 75, the authors should give brief detail on how to produce the anti-polypeptide (140-160 of VP1) monoclonal antibody or provide the references.

7. Line 77, recombinant protein(s) and plasmid DNA map(s) (with a label of each constructs) should be provided as a figure in the manuscript or the supplementary data.

8. Line 80, what is the PCR template used to amplify the G-H loop?

9. Line 81, is it OE-PCR or SOE-PCR? Please provide a reference.

10. Line 89, E. coil should be spelled out in full, when first mentioned.

11. Line 97 and 101, 'rpm' stands for 'round per minute' already. No need for '/min'. Actually, if the authors could provide a centrifugation rate in term of 'x g', it will be more useful for reproducing the experiments.

12. Line 118, please provide additional information on how to maintain and inactivate FMDV, or at least provide a reference.

13. Line 120 and 132, where the pig anti-FMDV/O hyper immune serum and the rabbit anti-FMDV serum came from? (commercialized source? reference?)

14. Line 130, did pre-immune serum samples from mice were included in the experiments?

15. Line 135, why did not coat the inactivated virus directly onto the 96-well plate?

16. In the Methods section, please provide a source of each commercial product, such as, Line 137, goat anti-mouse IgG-HRP.

17. Line 163, 's,' ?

18. Line 166, what bands are expected for the RFLP?

19. In general, Figure legend should include adequate information for its described figure, without going back to the text. Markers should be used to indicate the bands in the gel. For example, in Figure 1, the authors should indicate the size of each band. In the legend for Figure 1B, what restriction enzymes were used? In Fig 1A, cannot see the expected band in Lane 3.

20. Figure 5-7, the use of asterisks (*, **, ****) are very confusing, please revise accordingly. In the Figure legend and table description, if possible, use of abbreviations should be avoid, for example, in Figure 6, CoA, dpv, and FMDV.

---

## Round 0.2 · accepted · Accept

The manuscript has been amended according to the referees' comments

# Reviewer 2 ·

Basic reporting

-

Experimental design

-

Validity of the findings

-

Additional comments

A minor correction: Escherichia coli, not Escherichia Coli